# Pharmacist-Driven Alcohol Use Disorder Screening May Increase Inpatient Utilization of Extended-Release Naltrexone: A Single Center Pilot Study

**DOI:** 10.3390/pharmacy12010026

**Published:** 2024-02-01

**Authors:** Sabrina Snyder, Niyati Butala, Andrew M. Williams, Jamie Kneebusch

**Affiliations:** 1Department of Pharmacy, Arlington Campus, Riverside University Health System, Riverside, CA 92503, USA; s.snyder@ruhealth.org (S.S.); n.butala@ruhealth.org (N.B.); andrew.williams@ruhealth.org (A.M.W.); 2Skaggs School of Pharmacy and Pharmaceutical Sciences, University of California, San Diego, CA 92093, USA

**Keywords:** alcohol use disorder, inpatient, mental health, naltrexone, patient readmission, pharmacist

## Abstract

Individuals with mental illness have a high incidence of comorbid substance use, with one of the most prevalent being alcohol use disorder (AUD). Naltrexone, FDA-approved for AUD, decreases reward associated with alcohol-related social cues. This study aimed to determine if a pharmacist-driven screening tool would increase the use of extended-release naltrexone (XR-NTX) in patients with AUD and a comorbid psychiatric condition. Pharmacists screened and recommended XR-NTX for adults admitted to the inpatient psychiatric unit, who had a DSM-5 diagnosis of AUD, a negative urine drug screen for opioids, and were hospitalized for at least 1 day. Endpoints evaluated included the number of XR-NTX doses administered during the screening period to the prescreening period, 30-day readmission rates, recommendation acceptance rates, and reasons for not administering XR-NTX. Pharmacists identified 66 of 641 screened patients who met the inclusion criteria and were candidates for XR-NTX. Compared to the preintervention period, more patients received XR-NTX for AUD (2 vs. 8). Readmission rates were similar between those with AUD who received XR-NTX and those who did not. Pharmacist-driven screening for AUD led to greater administration of XR-NTX when compared to the same 4-month period the year prior to initiating the study.

## 1. Introduction

Alcohol use disorder (AUD) is a problem that affects millions of people throughout the United States. This disorder leads to billions of dollars in medical and criminal justice costs and lost productivity, and it is associated with an increased rate of morbidity and mortality [1,2]. People with mental health disorders have a high incidence of comorbid substance use disorders, including AUD. A 2017 study found that 21% of patients 12 years and older with a mental health condition also had a diagnosis of AUD [3]. Riverside County, located in California, has experienced a steady increase in the rate of hospitalizations due to alcohol use. Between 2016 and 2019 in Riverside County, 9.2% of all inpatient hospital admissions were related to alcohol, with a 4.8% increase in alcohol-related hospitalizations over that same period. Between 2019 and 2021, 23.8% of alcohol-related emergency department visits involved a concomitant mental health disorder, suicidal ideation, or suicide attempt [4]. At Riverside University Health System (RUHS) Arlington campus, patients are managed for comorbid AUD in addition to their psychiatric illness. Excessive alcohol use is associated with medication non-adherence, poorer treatment outcomes, lack of symptom stabilization, rehospitalization, homelessness, and lower quality of life [5,6,7,8].

Naltrexone is an evidence-based medication that is FDA-approved for the treatment of AUD [9]. According to the American Psychiatric Association, naltrexone and acamprosate should be offered to patients with moderate to severe AUD who have a goal of reducing alcohol consumption or achieving abstinence [10]. According to the COMBINE study, naltrexone plus medical management led to a greater proportion of days abstinent and a reduced risk of having a heavy drinking day, whereas acamprosate did not have the same benefit for drinking outcomes [11]. Thus, naltrexone should be considered as the first-line treatment option for patients with AUD, unless contraindicated.

Naltrexone acts as a competitive antagonist at the mu opioid receptor, and it is thought to reduce the reward associated with drinking alcohol or when encountering alcohol-related environmental cues [9,11]. The medication is available as a tablet, administered daily, and as a long-acting intramuscular injectable formulation, administered monthly. While both preparations effectively lengthen the time to first heavy drinking day among individuals with AUD, extended-release naltrexone (XR-NTX) has demonstrated greater adherence over the oral formulation [12,13]. In addition to decreasing heavy drinking, XR-NTX has been shown to be effective in reducing alcohol cravings, decreasing alcohol consumption during holiday periods, increasing rates of abstinence, and delaying time to relapse [1,14,15,16]. Following the first injection of XR-NTX, patients are reported to have 37% fewer heavy drinking days compared to placebo [17]. The onset of the therapeutic effect of XR-NTX has been observed within the first 2 days after the first injection, and the benefit of reduced heavy drinking days is sustained over several months [17]. In addition, patients who received XR-NTX had a lower 1-year mortality than those who did not receive treatment [18]. Despite the benefits of using XR-NTX for AUD, utilization rates remain low in the inpatient psychiatric setting. One potential barrier to use is the lack of knowledge and familiarity of physicians with prescribing XR-NTX [19].

At RUHS Arlington campus, usage of long-acting injectable (LAI) antipsychotics exceeds (22%) that currently reported nationally (1.1–10.3%) [20]. This increased utility at RUHS Arlington of LAIs was calculated based on the ratio of patients receiving a LAI compared to the total number of patients taking scheduled antipsychotics. This difference was determined following the implementation of a pharmacist-driven screening process in this inpatient mental health setting contributing to the increase in utilization by identifying and documenting patients who may benefit from a LAI medication. To address low utilization and prescribing barriers with XR-NTX at the RUHS Arlington campus, a pharmacist-driven screening of patients was implemented to identify those who met the criteria for treatment with XR-NTX for AUD. Data supporting the use of XR-NTX for alcohol dependence demonstrate that administration results in decreased inpatient and emergency healthcare costs and utilization [13,21]. The aim of this study was to determine if a pharmacist-driven screening tool would increase the utilization of XR-NTX in patients with AUD and a comorbid psychiatric condition. It is hypothesized that by screening patients hospitalized in the inpatient psychiatric setting for AUD, pharmacists’ interventions will lead to an increase in the administration of XR-NTX, and lead to a decrease in 30-day readmission rates.

## 2. Materials and Methods

Pharmacists developed a screening tool, based upon package insert criteria for the safe administration of XR-NTX, to identify patients who met the study criteria to receive treatment with XR-NTX for AUD. The screening tool incorporated three broad sections: 1. Patient identifying information including history of present illness, allergies, height and weight, pregnancy or lactation status, and substances present in a urine drug screen; 2. Evaluation of patient’s qualification to receive XR-NTX, based on package insert recommendations and contraindications; and 3. Pharmacist recommendation for XR-NTX based on patient assessment in part 2 [9] (Appendix A).

Pharmacists screened all patients admitted to the inpatient psychiatric unit at RUHS Arlington campus between 1 September 2019 and 31 December 2019. The psychiatric hospital is a county facility with segregated adolescent and adult units. There three adult units, two 28-bed general units each and a single, nine-bed intensive care unit, all managed by psychiatrists and psychiatry fellows.

Eligibility for this study and documented in the assessment area (part 1 and 2) of the screening tool, included being at least 18 years of age with a DSM-5 diagnosis of moderate-to-severe alcohol use disorder and being hospitalized in the inpatient psychiatric setting at the RUHS Arlington campus for at least 1 day. Patients were required to have a negative urine drug screen for opioids at the time of admission to the hospital. Exclusion criteria included opioid use within the past 7–10 days, opioid dependence or active withdrawal from opioids, acute hepatitis, transaminitis (AST/ALT > 3 times upper limit of normal), pregnancy, and severe renal impairment (eGFR < 30 mL/min) [9].

Upon admission to the inpatient psychiatric hospital, patients were screened to identify if inclusion criteria were satisfied. If patients were deemed candidates, the pharmacist notified the provider that the patient may benefit from treatment with XR-NTX for AUD. The pharmacist would document their recommendation for XR-NTX in the electronic medical record as a “progress note” in the patient’s chart, which served as provider notification for the purposes of this study. Once the patient was discharged from the hospital, the study investigators conducted a chart review to identify if XR-NTX was administered during the inpatient hospitalization. In addition, 30-day readmission rates were analyzed for all patients recommended to receive XR-NTX for AUD. Hospital readmission was verified by examining the electronic medical records of RUHS and other affiliate institutions utilizing the same electronic medical record. The number of XR-NTX doses administered during this study period was compared to a prescreening period between 1 September 2018 and 31 December 2018. Since pharmacists began recommending XR-NTX to providers at the beginning of 2019, study investigators decided to compare the number of doses administered during the screening period to the same 4-month period in the previous year.

Demographic and clinical data were collected using chart review. Baseline data included age, sex, ethnicity, weight, height, body mass index, and housing status. Clinical data included primary diagnoses, urine drug screen, pregnancy status, alcohol level, renal and hepatic function, and presence of methamphetamine dependence. Other data collected included the reason for not administering XR-NTX, if available, and discharge placement. All diagnoses were made in accordance with the Diagnostic and Statistical Manual of Mental Disorders, 5th edition (DSM-5) and the International Statistical Classification of Diseases and Related Health Problems, 10th revision (ICD-10) codes [22,23]. Safety and tolerability data were collected using provider and nursing progress notes and evaluation of administered medications used to treat side effects associated with the study medication. This study was approved by the Riverside University Health System Institutional Review Board on 31 October 2019. Prior to the administration of XR-NTX, patients were required to sign a medication consent form acknowledging their willingness to receive the study medication.

The primary endpoint was to compare the number of XR-NTX doses administered during the screening period with the prescreening period of all patients screened. Secondary endpoints were conducted as a sub-analysis of the identified group of patients who were recommended XR-NTX and included 30-day readmission, physician acceptance rates for administering XR-NTX, and reasons for not administering XR-NTX. One exploratory endpoint assessed included the impact on methamphetamine use of XR-NTX in those with comorbid methamphetamine use disorder.

## 3. Results

Overall, 641 patients were screened upon admission to the inpatient psychiatric hospital with 77 (12%) patients meeting the diagnostic criteria for AUD. Among those individuals, XR-NTX was recommended to 66 patients who met the inclusion criteria for this study. The most common reason that patients were excluded was the presence of comorbid opioid use disorder.

Table 1 shows the baseline characteristics of the patients included in this study. Most patients were white males, with a mean age of 37 years old. All patients diagnosed with AUD also had a dual mental health diagnosis. Upon discharge from the hospital, about 60.6% of patients had a primary psychotic disorder diagnosis while 39.4% were diagnosed with a mood disorder. Psychotic disorders included schizophrenia, schizoaffective disorder, and unspecified psychosis. Mood disorders comprised bipolar disorder, depression, anxiety, and unspecified mood disorder. In addition, more than half of the patients with AUD had a comorbid diagnosis of methamphetamine use disorder. Over one-third of patients admitted to the inpatient psychiatric hospital with AUD were homeless.

During the 4-month prescreening period, two patients received XR-NTX for AUD. Among patients with AUD who were actively screened and were recommended as candidates for XR-NTX, a total of eight patients received the medication during their hospitalization. Of patients who were in the active intervention group that received the study medication, the majority (75%) did not have any readmissions within 30 days of discharge (Figure 1). In comparison, a quarter of patients (2/8) were readmitted to the hospital for psychiatric reasons related to their primary mental health diagnosis. A smaller percentage of actively screened patients who did not receive XR-NTX were readmitted to the inpatient psychiatric hospital for a mental health reason (6/58). Similarly, most patients in the prescreening group were not readmitted within 30 days of discharge (81%). When compared to those who received XR-NTX, more patients in the prescreening group were readmitted for an alcohol-related reason (2/58) or for another medical indication (3/58).

The implemented screening method yielded a 36% provider acceptance (24/66). Provider acceptance was defined as the psychiatrist offering the study medication to the patient, resulting in either refusal of any form of naltrexone, accepting to receive oral naltrexone, or receiving the long-acting formulation. Of all the patients recommended XR-NTX (*n* = 66), 12.1% (*n* = 8) received the recommended agent, 10.6% (*n* = 7) opted for oral naltrexone and 13.6% (*n* = 9) refused either formulation. There were no clinically significant adverse drug events associated with treatment in individuals who received XR-NTX. Of those patients who were included but ultimately did not receive XR-NTX (*n* = 58), the primary reason was due to the psychiatrists’ clinical prerogative (Figure 2).

Upon discharge from the hospital, half of the patients who received XR-NTX were placed at a step-down facility pending rehabilitation placement, while 25% of patients were transferred directly to the substance use recovery center. In comparison, more individuals who chose to receive oral naltrexone were discharged to self/home (43%), and fewer patients decided to go to the rehabilitation facility (14%). Most patients who did not receive any naltrexone product chose to be discharged to self and opted not to go to a rehabilitation facility.

In this study, more than half of the patients in the screening group (41/66) had comorbid methamphetamine use disorder and AUD, but only six of those individuals received XR-NTX. Of those individuals, two patients were readmitted to the hospital for psychiatric reasons. One patient had a negative urine drug screen for amphetamines upon readmission while the second individual did not have a drug screen collected at the time of readmission.

## 4. Discussion

Based on the results from this study, implementation of a pharmacist-driven screening tool in the inpatient psychiatric setting may increase utilization of XR-NTX in patients with AUD and a comorbid psychiatric condition. This pilot study was based upon a similar pharmacist-driven screening protocol implemented within the institution to improve the use of LAI antipsychotics, which yielded an increase in LAI antipsychotic use from 13% to 21.57% [24]. Given the success seen from proactive pharmacist screening in the utility of LAI antipsychotics, this study was developed to expand upon interventions made by pharmacists within the institution and to assess if similar benefits would be conferred for agents like XR-NTX. It was found that 12% of adults with a psychiatric condition were hospitalized in the inpatient psychiatric setting with a dual diagnosis of AUD, which though numerically lower than the 21% assessed nationally, can be explained by the inclusion of adults alone versus those 12 and older the national data includes [3]. Ultimately, following the implementation of pharmacist-based screenings, administration of XR-NTX increased numerically, from two in the pre-screening cohort to eight administrations in the active screening cohort, which though small in overall number, conveys similar trends as seen in the LAI antipsychotics screening protocol [24].

In this study, psychotic disorders and mood disorders were the primary mental health diagnoses among patients with AUD. For patients with a mental health diagnosis, substance use disorders are among the greatest barriers to functional recovery, especially in patients with schizophrenia. Individuals with schizophrenia have a three times higher likelihood of having AUD compared to the general population [25,26]. Several trials have demonstrated that naltrexone is effective in reducing the number of heavy drinking days as well as increasing the time in abstinence compared to placebo in patients with schizophrenia [27]. In an 8-week open-label study, naltrexone was given to patients with schizophrenia-spectrum and comorbid AUD three days per week. The results demonstrated significant reductions in addiction severity, alcohol cravings, number of drinks per week, drinks per drinking day, and days of drinking to intoxication [27]. Furthermore, naltrexone has been found to be the most effective anti-craving agent for individuals with comorbid substance use disorder and schizophrenia or bipolar disorders [28]. Similarly, in patients with bipolar I and II disorder, naltrexone resulted in reduced alcohol consumption and improved mood [29,30]. The number of patients in this study precluded investigators from making definitive comparisons to previous literature. Despite the small sample size, trends showed reductions in admissions secondary to alcohol use in those who received XR-NTX, diagnosed with psychotic and mood disorders, which is consistent with results seen in the comorbid psychiatric population of previous studies. Nevertheless, larger patient numbers would be needed to make true comparisons of this study to other available literature.

Contrary to the study hypothesis that increased use of XR-NTX would reduce hospital readmissions, readmission rates for any cause were similar between those individuals who received XR-NTX and those who did not. The percentage of patients who were not readmitted within 30 days of discharge was also similar between groups. However, patients with AUD who did not receive XR-NTX had a numerically higher rate of readmission due to alcohol than patients who received the study medication. Among those individuals who did not receive XR-NTX, we further investigated if the decision was made primarily by the psychiatrist or by the patient. For most patients, providers did not offer XR-NTX to the patients either due to an oversight or focusing more on the primary psychiatric diagnosis for which the patient was hospitalized. In addition, the potential lack of provider knowledge regarding XR-NTX and the benefits of utilizing the medication in this patient population may have contributed to low prescribing rates. When psychiatrists did recommend XR-NTX, there was a similar rate of patient acceptance between the long-acting injection and the oral formulation. It is possible that patients who refused both the oral and long-acting naltrexone did not feel adequately prepared to address their alcoholism during their hospitalization.

Study investigators also reviewed the number of patients with comorbid methamphetamine use disorder and the potential effect that XR-NTX would have on substance use as an exploratory outcome. A 2021 randomized controlled trial showed XR-NTX was effective in promoting abstinence from methamphetamine when administered every 3 weeks and in combination with bupropion [31]. However, literature remains inconclusive to support the use of XR-NTX alone in patients with stimulant use disorders despite evidence suggesting that naltrexone attenuates craving levels and the subjective effects of amphetamines [32,33]. One-third (2/6) of patients in this study with comorbid methamphetamine use disorder and AUD who received XR-NTX were readmitted. Neither patient had a positive drug screen during their readmission. While overall the outcomes in this population were positive, the small sample and exploratory nature of this outcome limits the determination of a causal relationship of XR-NTX on methamphetamine use. Further investigation may be warranted into this comorbid population in future studies.

There are several limitations to consider in this study. First, the small sample size limits the generalizability of the results to other patients with a dual diagnosis of AUD and a psychiatric condition, as well as broader statistical analysis. In addition, the disproportionate number of patients between the XR-NTX and the control groups may have resulted in inadvertently skewed data. One contributing factor to the limited number of patients was the short study duration. By increasing the screening duration, a larger number of patients could be sampled, which would allow for equal distribution between the groups, and thus a more representative outcome assessment. Additionally, the inclusion of individuals in the pre-screening period to include those who did not receive XR-NTX would make for a more representative comparator sample, and thus better generalizability. This improvement would allow for evaluation of whether increased XR-NTX use was secondary to interventions or from a natural increase in AUD-related admissions. Furthermore, the lack of provider knowledge and education surrounding how to locate pharmacist screening recommendations may have limited the prescribing of XR-NTX for patients with AUD. In the future, surveying provider hesitancy to prescribing will aid in directing appropriate education and potentially increasing psychiatrist XR-NTX prescribing rates. Recidivism rates were assessed using the institution’s electronic medical record, which allows users to view if a patient has been hospitalized at any other facility that utilizes the same system. However, if a patient was readmitted within 30 days to an outside hospital that does not use the same electronic medical record, their readmission information could not be retrieved, thus limiting the strength of the data and subsequent analysis of the outcome. This limitation also prevented the evaluation of ongoing use or discontinuation of XR-NTX in patients who received it while hospitalized.

## 5. Conclusions

Pharmacist-driven screening of patients admitted to the inpatient psychiatric hospital numerically increased the utilization of XR-NTX in individuals with comorbid AUD and a mental health diagnosis. Readmission rates were similar between those individuals with AUD who received XR-NTX and those who did not. Future studies would benefit from the inclusion of a broader prescreening cohort, a collection of reasons why XR-NTX was not chosen by providers despite pharmacist recommendation, and if necessary, pre-education of providers on the benefits of XR-NTX to allow for higher utilization of medications for AUD.

This single institution pilot study relays the potential benefit of implementation of a pharmacist-driven screening of hospitalized patients for appropriateness to receive XR-NTX. Despite the limitations of this study, it should not deter other facilities from trialing a similar screening tool within their own patient populations, as the screening tool was mirrored on FDA labeling and assessment of patients for medication appropriateness falls in line with the roles and responsibilities of pharmacists. This proactive tool may lead to benefits for the provider and patients, by evaluating patients for medications that may not have been considered but patients are eligible to receive. The tool can be adapted to any institution’s formulary and need not be limited to long-acting injectables. It can be formally published within an electronic medical record or shared with providers on rounds for their consideration if electronic medical record publication does not fall within pharmacist scope at other institutions.

## Figures and Tables

**Figure 1 pharmacy-12-00026-f001:**
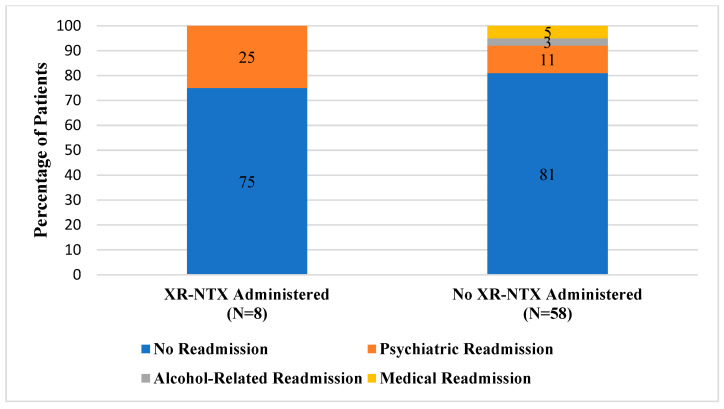
Comparison of 30-Day Readmission Rates.

**Figure 2 pharmacy-12-00026-f002:**
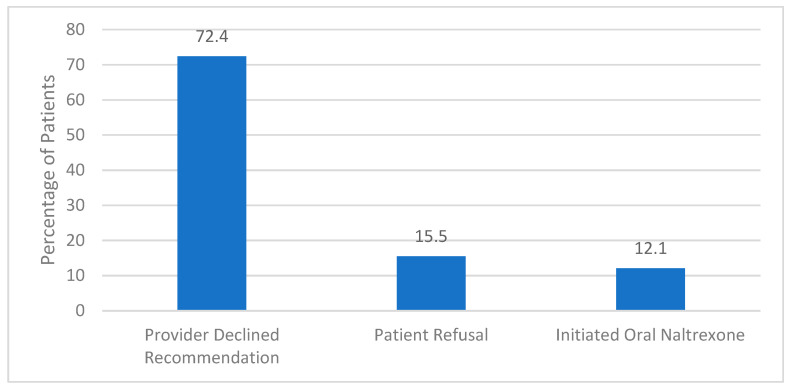
Reason for Not Receiving Extended-Release Naltrexone (*n* = 58).

**Table 1 pharmacy-12-00026-t001:** Clinical and Demographic Characteristics of Patients.

Characteristic	Pre-Screening Patients Receiving Extended-Release Naltrexone (*n* = 2)	Extended-Release Naltrexone Recommended (*n* = 66)	*p*-Value
Age–year	39.0 ± 8.5	36.6 ± 11.6	0.771
Male sex—*n* (%)	1 (50)	53 (80.3)	0.372
Race or ethnic group—*n* (%)			
White	1 (50)	30 (45.5)	0.971
Black	0	6 (9.1)	
Hispanic	1 (50)	29 (43.9)	
Other	0	1 (1.5)	
Comorbid Methamphetamine Use Disorder—*n* (%)	2 (100)	41 (62.1)	0.533
Primary Diagnosis—*n* (%)			
Psychotic disorders	2 (100)	40 (60.6)	0.521
Housing Status—*n* (%)			
Homeless	1 (50)	25 (37.9)	1.0

## Data Availability

The data are not publicly available due to being of or related to mental health care and the protection of patient privacy.

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
