# Peer review of "Pharmacist-Driven Alcohol Use Disorder Screening May Increase Inpatient Utilization of Extended-Release Naltrexone: A Single Center Pilot Study"

_pharmacy, 2024, doi:10.3390/pharmacy12010026_

Round 1

Reviewer 1 Report

Comments and Suggestions for Authors

The research offers informative insights into the potential effectiveness of a pharmacy screening for potential extended-release naltrexone for individuals with an alcohol use disorder (AUD) in an inpatient psychiatric unit. The authors provide an excellent overview of the effectiveness of XR-NTX to treat AUD and how the pilot study was carried out in the inpatient psychiatric unit. However, additional details or potential descriptions in the manuscript would enhance its impact in the field.

The authors discuss the pharmacy screen tool which as described starting in line 91 appears to follow the packaging recommendations for XR-NTX without any direct indication of how the materials may have been adapted for use in the screening tool. If the tool is solely based on the package insert, this should be clarified. If not, additional details about the screening tool and its component should be included in the manuscript. It may also be of value to include a screen shot of the screening tool as an online appendix.

The authors starting around line 101 discuss the process for pushing the results of the screening to the provider. However, it is not clear what that looks like in the electronic medical record. Specifically, how are the providers being notified of the results of the screening? How do they access the results in the electronic medical record. Again, it may be of value to include a screen shot of the push notification process in the electronic medical record as an online appendix.

In line 172, the authors state that 11% of patients who did not want to receive XR-NTX initiated oral naltrexone. However, the information in Figure 2 places a value of 12.1% for oral naltrexone initiation. Which number is correct? It is also not clear if the percentages in Table 2 are based on the 42 instances when the provider did not accept the screening recommendation or some other value.

The sentence starting in line 167 regarding provider acceptance (24 of 66) and the 12% receiving XR-NTX is confusing as written. It appears that acceptance is related to provider agreement of the pharmacist screening recommendation.

In line 186, the authors state that half of the patients had a comorbid methamphetamine use disorder. However, it is not clear what in the patient sample is half of? Please clarify.

In the discussion on line 238, the authors state that “contrary to study hypothesis” prior to discussing the findings related to readmission rates. However, the article did not clearly state any study hypothesis. Please explain.

The statement in the discussion starting with “nevertheless” in line 234 seems to be a rather bold statement given that the sample of patients who received XR-NTX was only 8 individuals. Further information is needed to back up this statement or it should be revised.

Starting in line 253, the authors identify different limitations of the study. However, they do not indicate if lack of provider education about the screening or how to look for the pushed recommendation in the electronic health record may also have impacted the results. Should these be considered limitations. If so, please explain. If not, please provide more information as to why these should not be considered study limitations.

Author Response

The research offers informative insights into the potential effectiveness of a pharmacy screening for potential extended-release naltrexone for individuals with an alcohol use disorder (AUD) in an inpatient psychiatric unit. The authors provide an excellent overview of the effectiveness of XR-NTX to treat AUD and how the pilot study was carried out in the inpatient psychiatric unit. However, additional details or potential descriptions in the manuscript would enhance its impact in the field.

Thank you.

The authors discuss the pharmacy screen tool which as described starting in line 91 appears to follow the packaging recommendations for XR-NTX without any direct indication of how the materials may have been adapted for use in the screening tool. If the tool is solely based on the package insert, this should be clarified. If not, additional details about the screening tool and its component should be included in the manuscript. It may also be of value to include a screen shot of the screening tool as an online appendix.

Added screenshot of screening tool to appendix.

The authors starting around line 101 discuss the process for pushing the results of the screening to the provider. However, it is not clear what that looks like in the electronic medical record. Specifically, how are the providers being notified of the results of the screening? How do they access the results in the electronic medical record. Again, it may be of value to include a screen shot of the push notification process in the electronic medical record as an online appendix.

Line 101 indicates how providers were informed via EMR progress note documentation. Added wording to indicate this is the formal provider notification.

In line 172, the authors state that 11% of patients who did not want to receive XR-NTX initiated oral naltrexone. However, the information in Figure 2 places a value of 12.1% for oral naltrexone initiation. Which number is correct? It is also not clear if the percentages in Table 2 are based on the 42 instances when the provider did not accept the screening recommendation or some other value.

Both numbers are correct and are conveying different percentages. 11% reflects the percentage who received PO naltrexone of those recommended the agent. Adjusted the verbiage and organization in this section to reflect what the percentages are conveying more clearly. Also updated Figure 2 descriptor to convey the number of patients being reflected in the percentages, n=58, which includes those who did not receive the drug of interest.

The sentence starting in line 167 regarding provider acceptance (24 of 66) and the 12% receiving XR-NTX is confusing as written. It appears that acceptance is related to provider agreement of the pharmacist screening recommendation.

Your understanding is correct. I adjusted organization of this paragraph to more clearly convey percentages and what values are being reported.

In line 186, the authors state that half of the patients had a comorbid methamphetamine use disorder. However, it is not clear what in the patient sample is half of? Please clarify.

Added n and divisor of total number of patients recommended to clarify.

In the discussion on line 238, the authors state that “contrary to study hypothesis” prior to discussing the findings related to readmission rates. However, the article did not clearly state any study hypothesis. Please explain.

Added a clarification of our hypothesis surrounding the readmissions.

The statement in the discussion starting with “nevertheless” in line 234 seems to be a rather bold statement given that the sample of patients who received XR-NTX was only 8 individuals. Further information is needed to back up this statement or it should be revised.

Revised the language to make the statement less bold and discussed the trends of our data being in line with other literature. Emphasized that larger patient numbers would be needed to make true comparisons.

Starting in line 253, the authors identify different limitations of the study. However, they do not indicate if lack of provider education about the screening or how to look for the pushed recommendation in the electronic health record may also have impacted the results. Should these be considered limitations. If so, please explain. If not, please provide more information as to why these should not be considered study limitations.

Thank you for this recommendation. I agree that this is a limitation and have included additional language that provider education on locating the pharmacist’s recommendation would be helpful for future studies.

Reviewer 2 Report

Comments and Suggestions for Authors

Thanks for allowing me to contribute as a reviewer. This is an interesting study that I think will be of interest to the journal's readership. Writing structure can be improve to be more concise and to the point at times. 

There are a few logical flaws in the methods, the presentation of the data in Table 1, and failure to mention key confounders of the results under study. 

Pg 1 Lines 33-35: suggest revision to "...high incidence of comorbid substance use disorders, including AUD. A 2017 study found that 21% of patients diagnosed with mental health condition also had a diagnosis of AUD."

line 36: suggest replacing "people hospitalized" with "hospitalization"...suggest mentioning what State Riverside County is located in. 

Line 45 suggest revision to "Naltrexone is an evidence-based medication that is FDA-approved for the treatment of AUD."

Line 55-56: Vivitrol is an intramuscular injection (not subcutaneous) please correct this. 

Line 70-71: This is interesting, but requires more clarification about how this was calculated. The reader is left wondering how this was measured. 

Line 99: suggest revision to "If patients were deemed candidates, the pharmacist notified the provider that the patient may benefit from treatment with XR-NTX for AUD."

Methodology: If you are looking at readmission rates you should likely collect and report the number of prior psychiatric hospitalizations for each patient. 

Line 139: Table 1 actually only includes data about the patients you recommended XR-NTX for, not the patients included in this study which would be everyone you screened and ideally the historic comparison group you looked at. Since your primary endpoint is doses administered in the screening vs pre-screening period it'd be helpful to get an idea of what the patients in both groups look like separately. 

It seems as though you may have not examined an actual historic group and just looked at XR-NTX utilization in the pre-screening period. 

Table 1 should provide the demographic and clinical data listed (and number of previous psych hospitalizations if available) for all the patients screened ( and the historic cohort if available). That way, the reader can compare the inferred natural history between the two groups and see if there is any significant between group differences that might confound the findings. To further describe what this should look like: Leave column 1, Replace column 2 with data from the 641 patients the Rph screened in 2019 and then add column 3 with data from the 2018 comparison group. There should also be 4th column for P-values ( use fisher's exact test or other suitable test) to determine if there are statistically significant differences in demographics and clinical characteristics between the two groups examined. If there are, you'll need to mention this in the limitations section of the manuscript b/c this contributes to confounding.

I suggest you make it a little more clear that the secondary endpoint in this study involves a sub-analysis of the current screening group. Not the same comparison group used in the primary endpoint. 

For the sake of interpretation of this data on secondary endpoint  it would also be helpful to use another table to describe demographic data and clinical characteristics for these patients and repeat what was done in table 1.  Some of what you have in table 1 could be reused for this purpose. Just add a column and display data for patients that didn't receive XR-NTX and then add another column for p-values. 

line 180: "Discharged to self" what does this mean?

Line 192: be more reserved in stating your conclusion: "may increase utilization of XR-NTX"

Line 216: replace with something like "Further investigation may be warranted". 

Section about XR-NTX for MUD should be revised and made to be much more concise. Not sure how deep into discussion about  XR-NTX's efficacy in MUD you want to get since you only had 6 patients with MUD, but if you're discussing this you should definitely mention that XR-NTX is effective at promoting abstinence from methamphetamine when given Q3weeks in combination with bupropion. There is NEJM medicine RCT about this. This reference could probably replace the others you have utilized in this section. 

There is a significant confounding factor that is not mentioned in the limitations section of this study and was alluded to in the introduction.

The rates of alcohol-related hospitalization are increasing at this hospital and jump sharply at the time this study was conducted (in 2019) see lines 35-41.

So any increase in the utilization of XR-NTX AUD that you observed in your study could also partly be due to fact that you're seeing a ton more AUD-related admissions during this time period. 

Updating the two tables  will greatly assist readers in assessing the likelihood of confounding present in the study. If you are able please also add the data I  requested. 

Key suggestions for improvement: Modify Table 1 to better allow for comparison of the data between groups for the primary outcome (see above)

Add a Table 2: present demographic data and clinical characteristics for patients you recommended XR-NTX for n=66. Column 1 should be for those that received the drug, column 2 for those that didn't column 3 for p-values. This should come before Figure 1 and will allow for better interpretation of data for secondary outcome. 

Eliminate the current Table 2. It's not a focus of discussion in the manuscript and doesn't offer much value to the reader. 

Add more description into Figure 2 next to "Provider" label in the graph. It is currently unclear what this means and there is no footnote to inform the reader. 

I believe this manuscript is close to publication it just requires some minor revisions.  I encourage the authors to think about how to improve the presentation of data in tables 1 and 2 to better allow readers to interpret there findings. 

Comments on the Quality of English Language

English is acceptable and requires only minor revisions throughout. 

Author Response

Thanks for allowing me to contribute as a reviewer. This is an interesting study that I think will be of interest to the journal's readership. Writing structure can be improve to be more concise and to the point at times. 

Thank you for serving as a review and your contribution to improving this manuscript.

There are a few logical flaws in the methods, the presentation of the data in Table 1, and failure to mention key confounders of the results under study. 

Will take the suggestions henceforth under advisement.

Pg 1 Lines 33-35: suggest revision to "...high incidence of comorbid substance use disorders, including AUD. A 2017 study found that 21% of patients diagnosed with mental health condition also had a diagnosis of AUD."

Suggestion incorporated while including ages of patients included in this study.

line 36: suggest replacing "people hospitalized" with "hospitalization"...suggest mentioning what State Riverside County is located in. 

Done.

Line 45 suggest revision to "Naltrexone is an evidence-based medication that is FDA-approved for the treatment of AUD."

Done.

Line 55-56: Vivitrol is an intramuscular injection (not subcutaneous) please correct this. 

Thank you for catching this. Revised.

Line 70-71: This is interesting, but requires more clarification about how this was calculated. The reader is left wondering how this was measured. 

Added a description of how the 22% value was calculated.

Line 99: suggest revision to "If patients were deemed candidates, the pharmacist notified the provider that the patient may benefit from treatment with XR-NTX for AUD."

Thank you. Updated to your suggested phrasing.

Methodology: If you are looking at readmission rates you should likely collect and report the number of prior psychiatric hospitalizations for each patient. 

Thank you for this suggestion. Given this was a prospective observational study, we were primarily interested in if the intervention conferred benefit to future readmissions. However, this is something that should be considered as baseline demographics for future studies. 

Line 139: Table 1 actually only includes data about the patients you recommended XR-NTX for, not the patients included in this study which would be everyone you screened and ideally the historic comparison group you looked at. Since your primary endpoint is doses administered in the screening vs pre-screening period it'd be helpful to get an idea of what the patients in both groups look like separately. 

Included the 2 patients from the pre-screening time period who received XR-NTX for comparison. We did not have the data from the pre-screening period of all patients from that time, only those who received NR-NTX, and so will keep table 1 as is for only those recommended in the active intervention time period.

It seems as though you may have not examined an actual historic group and just looked at XR-NTX utilization in the pre-screening period. 

Correct.

Table 1 should provide the demographic and clinical data listed (and number of previous psych hospitalizations if available) for all the patients screened ( and the historic cohort if available). That way, the reader can compare the inferred natural history between the two groups and see if there is any significant between group differences that might confound the findings. To further describe what this should look like: Leave column 1, Replace column 2 with data from the 641 patients the Rph screened in 2019 and then add column 3 with data from the 2018 comparison group. There should also be 4th column for P-values ( use fisher's exact test or other suitable test) to determine if there are statistically significant differences in demographics and clinical characteristics between the two groups examined. If there are, you'll need to mention this in the limitations section of the manuscript b/c this contributes to confounding.

Unfortunately, previous psychiatric hospitalizations were not collected for this study. Will keep in mind for future studies. Added information for the historic cohort, which only included those who received XR-NTX. Given the prescreening group only included those which received XR-NTX, felt it appropriate to keep the comparator group in the table to those who were appropriate to receive XR-NTX in the screened group.

I suggest you make it a little more clear that the secondary endpoint in this study involves a sub-analysis of the current screening group. Not the same comparison group used in the primary endpoint. 

Added to methods.

For the sake of interpretation of this data on secondary endpoint  it would also be helpful to use another table to describe demographic data and clinical characteristics for these patients and repeat what was done in table 1.  Some of what you have in table 1 could be reused for this purpose. Just add a column and display data for patients that didn't receive XR-NTX and then add another column for p-values. 

As above, I chose to keep the comparator group in table 1 as the 66 who were recommended NR-NTX rather than the entirety of the 641 patients screened to be in line with the data collected from the pre-screened group. Agree this would have allowed for more robust comparison and will keep in mind for future studies.

line 180: "Discharged to self" what does this mean?

It is referring to not being discharged to a facility (i.e. rehabilitation center, SNF, etc). Given many of our patients are unhoused individuals, discharge to home is not always inclusive of their discharge plan. Adjusted wording to be in line with Table 2, “discharged to self/home”

Line 192: be more reserved in stating your conclusion: "may increase utilization of XR-NTX"

Updated. Thank you.

Line 216: replace with something like "Further investigation may be warranted". 

Updated.

Section about XR-NTX for MUD should be revised and made to be much more concise. Not sure how deep into discussion about  XR-NTX's efficacy in MUD you want to get since you only had 6 patients with MUD, but if you're discussing this you should definitely mention that XR-NTX is effective at promoting abstinence from methamphetamine when given Q3weeks in combination with bupropion. There is NEJM medicine RCT about this. This reference could probably replace the others you have utilized in this section. 

Agreed. Incorporated the NEJM naltrexone-bupropion trial into this section and cut back on excessive language, given the exploratory nature of this outcome.

There is a significant confounding factor that is not mentioned in the limitations section of this study and was alluded to in the introduction.

Nothing to comment on here.

The rates of alcohol-related hospitalization are increasing at this hospital and jump sharply at the time this study was conducted (in 2019) see lines 35-41.

True.

So any increase in the utilization of XR-NTX AUD that you observed in your study could also partly be due to fact that you're seeing a ton more AUD-related admissions during this time period. 

Added as a limitation.

Updating the two tables  will greatly assist readers in assessing the likelihood of confounding present in the study. If you are able please also add the data I  requested. 

Addressed above, and below.

Key suggestions for improvement: Modify Table 1 to better allow for comparison of the data between groups for the primary outcome (see above)

Added pre-screening comparator group which was limited to those who received XR-NTX alone. Given this limitation of the pre-screened group, did not see the utility in providing broader screened group demographics, as it would not offer additional comparison opportunity.

Add a Table 2: present demographic data and clinical characteristics for patients you recommended XR-NTX for n=66. Column 1 should be for those that received the drug, column 2 for those that didn't column 3 for p-values. This should come before Figure 1 and will allow for better interpretation of data for secondary outcome. 

As above, given authors chose to keep table 1 as the demographics of those individuals who had active interventions, no need for additional table with same demographic information. I take your note however, that a limitation of this study was a lack of a robust prescreening process to allow for comparisons with the larger screened group.

Eliminate the current Table 2. It's not a focus of discussion in the manuscript and doesn't offer much value to the reader. 

Agreed. Removed the table, though kept the information in text format within the results section.

Add more description into Figure 2 next to "Provider" label in the graph. It is currently unclear what this means and there is no footnote to inform the reader. 

Thank you for catching this. Updated figure to provide better axis descriptors.

I believe this manuscript is close to publication it just requires some minor revisions.  I encourage the authors to think about how to improve the presentation of data in tables 1 and 2 to better allow readers to interpret there findings. 

Thank you for your suggestions.

Reviewer 3 Report

Comments and Suggestions for Authors

This is a cross-sectional pre-/post study of consecutive patients admitted to a psychiatric hospital in the last third of 2019 for inpatient treatment compared to 2018 use patterns and the effect of a pharmacist screening tool on the frequency of provider-prescribed injectable naltrexone administrations. Because of the very limited number of patients that agreed to the injection (8/66 - 12%), this would be at best called a pilot study. The manuscript is well written and appropriately referenced (although not in MDPI style). The following comments are offered in order to strengthen the presentation.

Title: long but descriptive. Consider dropping "in the in-patient psychiatric setting" and add: A Single Center Pilot Study. Thus, "Pharmacist-Driven Alcohol Use Disorder Screening May Increase Inpatient Utilization of Extended-Release Naltrexone: A Single Center Pilot Study."

Abstract: at line 16, adults are 18 years and older, so drop the "who were 18 years and older."

Keywords: please alphabetize them.

Introduction: a decent review of the literature.

At line 56, the authors state that the injection is subcutaneous. It is not according to the most recent labeling and is a deep intramuscular gluteal injection.

At line 74, the authors state data exists from another Riverside University Hospital, but they don't include the statistical results that would serve as the basis for their study in terms of estimated percentage change and variance. Please include if available.

Methods:

Why was a 4-month study period selected? 

At line 85, the authors mention that a screening tool was developed by the pharmacists. A copy of the screening tool should be available as a Supplementary Material. From what resources was the tool derived? Was it paper-based or entered into the EHR or CPOE system?

Results:

These injections are painful. I would like to see data on all patients that received follow up injections over the 4-month period, if available. If not, this is a limitation of the study design and should be stated thusly. Studies suggest that adherence is better with IM versus PO naltrexone.

How many pharmacists were involved in the screening process? What are their characteristics? Were any board-certified psychiatric pharmacists?

At line 170, the authors state that "psychiatric prerogative" was the primary reason for not prescribing IM naltrexone, and they conclude that psychiatrist education would improve uptake. What does this mean? Were there multiple psychiatrists involved? Were they able to administer an IM injection? You shouldn't design an educational intervention if you don't know why, Maybe the psychiatrists knew that the injection would be painful and didn't recommend it on that basis.

Discussion:

Why was it important that a pharmacist did the screening? How do you think the results would change if another provider administered the screening?

At lines 192-201, the results of the study are restated. The discussion should place the study's results in the context of the literature. What is unique about your study? What do your results mean? What might you do next in implementing the screening tool among pharmacists?

Limitations are reasonably explained with the addition of the lack of follow up data on repeat injections.

Conclusion: 

Can you speculate on what future research is needed to implement the screening tool at this and other facilities? 

Author Response

This is a cross-sectional pre-/post study of consecutive patients admitted to a psychiatric hospital in the last third of 2019 for inpatient treatment compared to 2018 use patterns and the effect of a pharmacist screening tool on the frequency of provider-prescribed injectable naltrexone administrations. Because of the very limited number of patients that agreed to the injection (8/66 - 12%), this would be at best called a pilot study. The manuscript is well written and appropriately referenced (although not in MDPI style). The following comments are offered in order to strengthen the presentation.

Thank you for your review and contribution to improving this manuscript.

Title: long but descriptive. Consider dropping "in the in-patient psychiatric setting" and add: A Single Center Pilot Study. Thus, "Pharmacist-Driven Alcohol Use Disorder Screening May Increase Inpatient Utilization of Extended-Release Naltrexone: A Single Center Pilot Study."

Thank you for this suggestion. Title has been updated.

Abstract: at line 16, adults are 18 years and older, so drop the "who were 18 years and older."

Removed.

Keywords: please alphabetize them.

Done

Introduction: a decent review of the literature.

Thank you.

At line 56, the authors state that the injection is subcutaneous. It is not according to the most recent labeling and is a deep intramuscular gluteal injection.

Thank you for catching this. Revised.

At line 74, the authors state data exists from another Riverside University Hospital, but they don't include the statistical results that would serve as the basis for their study in terms of estimated percentage change and variance. Please include if available.

This institution is the same institution at which the current study has been conducted. This was an unpublished study based on data at the institution. In line with another reviewer’s comments, I included how that value came to be calculated.

Methods:

Why was a 4-month study period selected? 

This was a resident research study and so the duration of time selected was due to the nature of a limited training time frame and requirements to complete data collection and analysis, time frame for the pilot study was limited to 4 months.

At line 85, the authors mention that a screening tool was developed by the pharmacists. A copy of the screening tool should be available as a Supplementary Material. From what resources was the tool derived? Was it paper-based or entered into the EHR or CPOE system?

Will include a copy of the screening tool as a supplement. The tool was derived from the package insert and recommendations were entered in the electronic medical record.

Results:

These injections are painful. I would like to see data on all patients that received follow up injections over the 4-month period, if available. If not, this is a limitation of the study design and should be stated thusly. Studies suggest that adherence is better with IM versus PO naltrexone.

Thank you for this perspective. This data was not collected and so will be included as a limitation and for consideration in future studies.

How many pharmacists were involved in the screening process? What are their characteristics? Were any board-certified psychiatric pharmacists?

This was resident run, and therefore was conducted by a single pharmacist who was not yet board certified.

At line 170, the authors state that "psychiatric prerogative" was the primary reason for not prescribing IM naltrexone, and they conclude that psychiatrist education would improve uptake. What does this mean? Were there multiple psychiatrists involved? Were they able to administer an IM injection? You shouldn't design an educational intervention if you don't know why, Maybe the psychiatrists knew that the injection would be painful and didn't recommend it on that basis.

The psychiatric hospital employed numerous psychiatrists, who all covered separate teams. On the adult units, we had upwards of 5 psychiatrists with their own unique clinical discretion. However, the point you make is fair, in that we did not gather more information as to why the recommendation was declined, and will address this limitation in the paper as a suggestion for inclusion in future studies.

Discussion:

Why was it important that a pharmacist did the screening? How do you think the results would change if another provider administered the screening?

Added context to the use of pharmacists conducting this screening, by relating back to data gathered at this same institution showing increased use of LAI antipsychotics which used a similar intervention method.

At lines 192-201, the results of the study are restated. The discussion should place the study's results in the context of the literature. What is unique about your study? What do your results mean? What might you do next in implementing the screening tool among pharmacists?

Updated data to place within the context of other literature and to provide context to the study.

Limitations are reasonably explained with the addition of the lack of follow up data on repeat injections.

Added.

Conclusion: 

Can you speculate on what future research is needed to implement the screening tool at this and other facilities? 

Added a short suggestion on how other institutions might add this screening tool to their formularies.

Round 2

Reviewer 1 Report

Comments and Suggestions for Authors

I appreciate the authors attention to detail in responding to reviewer comments.

Reviewer 3 Report

Comments and Suggestions for Authors

Thank you for addressing the reviewers' suggestions. I appreciate the opportunity to review your interesting work.